# Temporal discounting in adolescents and adults with Tourette syndrome

**Canan Beate Schüller**[1‡]*, **Ben Jonathan Wagner**[2‡], **Thomas Schüller**[1], **Juan Carlos Baldermann**[1,3], **Daniel Huys**[1], **Julia Kerner auch Koerner**[4,5], **Eva Niessen**[6], **Alexander Münchau**[7], **Valerie Brandt**[8], **Jan Peters**[2‡], **Jens Kuhn**[1,9‡]

**1** Department of Psychiatry and Psychotherapy, Faculty of Medicine and University Hospital Cologne, University of Cologne, Cologne, Germany, **2** Department of Biology Psychology, University of Cologne, Cologne, Germany, **3** Department of Neurology, Faculty of Medicine and University Hospital Cologne, University of Cologne, Cologne, Germany, **4** Educational Psychology, Helmut-Schmidt-University, Hamburg, Germany, **5** Center for Individual Development and Adaptive Education of Children at Risk, Frankfurt am Main, Germany, **6** Department of Individual Differences and Psychological Assessment, University of Cologne, Cologne, Germany, **7** Institute of Systems Motor Science, University of Lübeck, Lübeck, Germany, **8** Center for Innovation in Mental Health, School of Psychology, University of Southampton, Southampton, Hampshire, United Kingdom, **9** Department of Psychiatry, Psychotherapy and Psychosomatic, Johanniter Hospital Oberhausen, EVKLN, Oberhausen, Germany

‡ CBS and BJW share first authorship on this work. JP and JK share senior authorship on this work.
* canan.schueller1@uk-koeln.de

**Data Availability Statement:** We have now made the data publicly available on the platform: center for open science (Schüller C. Temporal discounting in adolescents and adults with Tourette syndrome

## Abstract

Tourette syndrome is a neurodevelopmental disorder associated with hyperactivity in dopaminergic networks. Dopaminergic hyperactivity in the basal ganglia has previously been linked to increased sensitivity to positive reinforcement and increases in choice impulsivity. In this study, we examine whether this extends to changes in temporal discounting, where impulsivity is operationalized as an increased preference for smaller-but-sooner over larger-but-later rewards. We assessed intertemporal choice in two studies including nineteen adolescents (age: mean[sd] = 14.21[±2.37], 13 male subjects) and twenty-five adult patients (age: mean[sd] = 29.88 [±9.03]; 19 male subjects) with Tourette syndrome and healthy age- and education matched controls. Computational modeling using exponential and hyperbolic discounting models via hierarchical Bayesian parameter estimation revealed reduced temporal discounting in adolescent patients, and no evidence for differences in adult patients. Results are discussed with respect to neural models of temporal discounting, dopaminergic alterations in Tourette syndrome and the developmental trajectory of temporal discounting. Specifically, adolescents might show attenuated discounting due to improved inhibitory functions that also affect choice impulsivity and/or the developmental trajectory of executive control functions. Future studies would benefit from a longitudinal approach to further elucidate the developmental trajectory of these effects.

## Introduction

Tourette syndrome (TS) is a childhood onset neuropsychiatric disorder characterized by motor and phonic tics that wax and wane in their severity with an estimated prevalence of

[Internet]. OSF; 10 Apr 2021. Available: osf.io/nw9pc).

**Funding:** C.B.S. and T.S. were funded by the Walter and Marga Boll Foundation (210-06-16, URL: https://www.bollstiftung.de/). T.S. and J.C.B. were supported by Deutsche Forschungsgemeinschaft (DFG: 431549029 – SFB 1451; URL: https://www.dfg.de/). J.P. was supported by Deutsche Forschungsgemeinschaft (DFG: PE 1627/5-1; URL: https://www.dfg.de/). A. M. was supported by the DFG (FOR 2698; URL: https://www.dfg.de/). V.B. was supported by the Academy of Medical Sciences (URL: https://acmedsci.ac.uk/). C.B.S., T.S., A.M., V.B, J.K., and J.P. conceived the idea. C.B.S., T.S., E.N., A.M., V. B., J.K.K., and J.P. conceived and designed the experiments; C.B.S., E.N., and V.B. performed the experiments; C.B.S., B.W., and J.P. analyzed the data; J.P. and B.W contributed analysis tools; B.W. performed the modelling. C.B.S. and B.W. wrote the paper. All authors reviewed and approved the final manuscript. The funders had no role in study design, data collection and analysis, decision to publish, or preparation of the manuscript.

**Competing interests:** The authors have declared that no competing interests exist.

around 1% [1]. Motor tics are repetitive, sudden movements such as eye blinking or facial muscle contractions and phonic tics are repetitive sounds such as throat clearing or verbal utterances [1, 2]. TS onset occurs predominantly in early childhood with a peak of symptom severity between the age of 10 and 12 years. Thereafter, tics improve in around 80% of children until the end of adolescence [3].

Both clinical and neuroscientific research have highlighted possible developmental dysfunctions in the cortico-striatal-thalamo-cortical (CSTC) loops [4–6] especially with respect to dopamine (DA) that strongly modulates these circuits [7, 8]. The striatum, a main gateway in these loops [9] plays a key role in selectively amplifying converging sensory input to enable situation specific behavioral adaptations, such as the adequate control of voluntary movement [7]. Predictions (i.e. expectations) of reward, as well as the gating of specific motor responses, are under dopaminergic modulation. Theories about the developmental underpinnings of TS in terms of DA function range from striatal DA receptor super-sensitivity [10] over tonic-phasic or presynaptic DA dysfunction [11, 12] to DA hyper-innervation [11, 13]. The DA hyper innervation hypothesis unifies previous findings under a promising framework.

To date, several studies have investigated motor impulsivity in patients with TS with reference to DA´s role in reward and motor control [14, 15]. However, fewer studies have explored alterations in value-based decision-making in TS. This question is of particular interest because motor and choice impulsivity might at least in part be supported by common neural systems. First, DA in fronto-striatal circuits plays a role in both motor control [16, 17] and choice impulsivity [18–20]. Second, some studies have suggested that lateral prefrontal cortical (LPFC) regions might support impulse control functions, both in motor and non-motor domains [21–24]. Two studies [25, 26] examined impairments in value-based decision-making in TS in the context of reinforcement learning tasks. Palminteri and Pessiglione observed impaired learning from negative feedback in TS [25], which is consistent with the idea of a hyperdopaminergic state. Kéri and colleagues observed impaired probabilistic classification learning, especially in children with severe tics [26]. However, the degree to which choice impulsivity is impaired in TS remains unclear. To date, only one study examined temporal discounting in patients with TS via the Kirby Monetary Choice Questionnaire [27] and observed slightly increased discounting compared to healthy controls.

Another way to reliably assess this process is via intemporal choice tasks [28, 29]. Temporal discounting describes a general preference for smaller sooner (SS) over larger, but later rewards (LL) [30]. A relative preference for SS rewards (steep discounting of value over time) is associated with a range of problematic behaviors, including substance use disorders and overweight/obesity [31], the tendency to procrastinate investing in a pension [32], and to procrastinate saving up for future investments [33]. The rate of temporal discounting is subject to complex modulation by individual and contextual variables [34, 35], where striatal DA networks and prefrontal top down modulation seem to play crucial roles. However, the precise relationship between dopaminergic states and impulsive choice is complex and might be a function of age with a proposed u-shape association [36]. On the one hand, pharmacological elevation of DA levels decreases discounting [20, 37–39]. On the other hand, hyperdopaminergic states, e.g. due to administration of the dopamine precursor L-DOPA, are in some studies associated with increased discounting [18], and patients with Parkinson's disease can exhibit increased impulsive behavior following DA replacement therapy [19]. To sum up, DA modulation likely contributes to the modulation of intertemporal choice via its action on different fronto-striatal loops, but scientific evidence suggests that there is no clear and simple linear relationship between DA levels and choice impulsivity.

The picture is clearer with regard to top-down inhibitory mechanisms. The LPFC is assumed to modify choice impulsivity [40–42], that is, inhibition of the selection of tempting

SS choices in this model depends on prefrontal inhibitory regulation of subcortical or ventro-medial prefrontal value representations. Changes in structural and functional connectivity within this network are linked to the development of self-control (in this study the term 'self-control' generally refers to far-sighted behavior in value based decision making) from adolescence to early adulthood [43, 44]. Furthermore, inhibition and top-down control play a central role in motor impulsivity and are believed to modulate TS pathophysiology, e.g. in the context of suppressing urges and tics [14].

Studies have shown that motor and cognitive impulsive actions might require different forms of self-control and these can be differentiated [45]. To sum up, there is extensive evidence for regional overlap between inhibitory mechanisms in terms of motor impulsivity, choice impulsivity, and other forms of impulsivity, such as emotion regulation [22–24]. Training in one domain might affect performance in other domains [46]. Regarding choice and motor impulsivity, the dorsal striatum might be a key region of interest where top down inhibitory processes (originating in the PFC) modulate the execution or the re-evaluation of choice outcomes [47]. These anatomical regions and attributed functions might be affected in TS pathophysiology [48]. Even though temporal discounting has been proposed as an transdiagnostic trait [49] with valuable diagnostic potential [50] it is still an open question whether patients with TS show aberrations in the domain of intertemporal choice. In the present study, we compared adolescents (Study 1, Hamburg) and adults (Study 2, Cologne) with TS to controls, using two modified temporal discounting tasks to broaden the understanding of value based decisions in TS on one operational measure of choice impulsivity [32, 33].

## Materials and methods

### Ethics

The ethics committee of the University Hospital Hamburg approved the first study. Adolescent patients with TS provided written assent and their parents provided written consent (PV4439). Patients with TS were recruited in the University Hospital of Hamburg, whereas controls were recruited by advertisement. The second study was carried out in accordance with institutional guidelines and was approved from the ethics committee of the University of Cologne (protocol ID: DRKS00011748). All participants provided written consent. Patients were recruited at the University Hospital of Cologne whereas controls were recruited by advertisement.

### Study 1 specific methods

**Participants.**   We included nineteen adolescents with TS (age: mean[sd] = 14.21[±2.37], 13 male subjects, range: 10–17) and nineteen age, education and gender-matched controls (age: mean[sd] = 14.21[±2.53], 15 male subjects, range: 10–18). Adolescents with TS were recruited from a specialist Tourette syndrome clinic in Hamburg. All patients had been diagnosed with Tourette syndrome, some had been treated for their tics. Healthy controls were partly recruited from a pool of healthy participants who had participated in a previous study, partly via public advertisement. All participants underwent a clinical assessment and performed a modified delay discounting paradigm. Two adolescents with TS were taking antido-paminergic drugs (Tiaprid) as prescription medication.

**Clinical assessment.**   Adolescents were assessed with the "Yale Global Tic Severity Scale" (YGTSS) [51], the "Premonitory Urge for Tic Disorders Scale" (PUTS), a self-report scale to identify premonitory urges [52], and the "Children's Yale-Brown Obsessive Compulsive Scale" (CY-BOCS), a semi structured interview to evaluate OCD severity. CY-BOCS data were available from all adolescents with TS and 13 controls; in total, three adolescents with TS had a

score above 12, the cut-off for clinically relevant OCD symptoms [53]. The "Parent-rated and Self-rated Questionnaires for Attention Deficit Hyperactivity Disorder" (German: "Fremd-beurteilungsbogen /Selbstbeurteilungsbogen für Aufmerksamkeitsdefizit-/Hyperaktivitätsstör-ungen") FBB-ADHD and SBB-ADHD are diagnostic instruments to identify ADHD [54]. FBB-ADHD data was available for all adolescents with TS and 16 controls. SBB-ADHS data was available for 18 adolescents with TS and 17 controls. All adolescents also filled out a questionnaire on demographic measurements (age, gender, medication).

**Task.** Participants performed a experiential delay discounting task based on prior procedure [55] where they chose between varying smaller sooner (SS $\epsilon$ [0, 1, 2, 3 or 4 cents]) or larger but later (LL [5 cents]) rewards. LL options were available after a specific waiting period of 10, 20, 30, 40 or 60 seconds. Each SS-option was paired twice with each LL-option resulting in 50 trials per participant. A progress bar indicated the number trials past. Position of the LL option was counterbalanced to the left or right side of the screen. LL waiting-time was visualized by the number of horizontal lines (e.g. 2 horizontal lines = 20s waiting period). Following choice rewards were transferred into a virtual piggy bank either immediately (if SS was chosen) or after the appointed waiting period (if LL was chosen). Depending on choices, participants could gain between 0 € and 2.50 €. Following this time spent with task, i.e. delay to reward delivery was related to the proportion of SS choices. (see **S1 Fig**).

## Study 2 specific methods

**Participants.** We recruited twenty-five patients with diagnosed TS according to DSM-5 criteria [56] from the psychiatric outpatient clinic of the University Hospital Cologne (age: mean[sd] = 29.88 [±9.03]; 19 male subjects, range: 19–53) and 25 age, education and gender-matched controls (age: mean[sd] = 29.40 [±9.28]; 17 male subjects, range:19–49) through public advertisement. All participants underwent a clinical assessment, performed a temporal discounting paradigm, including a pretest based on prior procedures [57, 58]. Nine patients were taking medication or cannabinoids. Five patients were treated with antidopaminergic drugs (Aripiprazole, risperidone, tiapride) as a monotherapy, one patient with an anticonvulsant (Valproate), one patient was taking a noradrenergic and specific serotonergic antidepressant (Mirtazapine), and one patient was medicated with a combination of two antidopaminergic drugs (Aripiprazole, risperidone) and a selective serotonin reuptake inhibitor (Citalopram). One patient regularly smoked medical cannabis.

**Clinical assessment.** All participants filled out the Obsessive Compulsive Inventory-Revised (OCI-R) [59] and the Beck Depression Inventory (BDI) [60]. The Wender Utah Rating Scale was used to assess ADHD symptoms [61]. Furthermore, they filled out a short intelligence test (Leitprüfsystem-3 (LPS 3)) [62], followed by a demographic questionnaire with information on age, gender, handedness, years of education and current drug or alcohol use. Further, patients with TS completed an assessment with the YGTTS [51], and the PUTS [52]. All questionnaires were in German.

**Task.** Prior to the first testing session, participants completed a short adaptive pretest to estimate the individual discount- rate ($k$). This discount rate was used to construct a set of 140 participant-specific trials using MATLAB (version 8.4.0. Natick, Massachusetts: The Math-Works Inc). The task consisted of choices between an immediate smaller-sooner reward of 20 € and participant specific larger-but-later (LL) rewards delivered after some delay (1, 2, 7, 14, 30, 90 or 180 days). In 70 trials, LL amounts were uniformly spaced between 20.5 € and 80 €, whereas in the remaining 70 trials LL amounts were uniformly spaced around each estimated indifference point per delay (based on the pre-test discount rate). If indifference points were larger than 80 €, only uniformly-spaced LL amounts were used. Trials were presented in a

pseudorandomized order. Participants were informed that after task completion, one trial would be randomly selected and paid immediately in cash (smaller-sooner choice) or via a timed bank transfer (larger-but-later choice).

## Statistical analyses (both studies)

**Model free analysis.**   Using model agnostic approaches can avoid possible caveats associated with model-based analysis, e.g., problems with parameter estimation or the choice for a theoretical framework. Due to task structure in study 1 (adolescents) we used the percentage of LL in contrast to SS choices as a model agnostic quantification of choice behavior. For comparison, we used a two-sided parametric test on the arc-sin-transformed values of SS vs. LL choices.

In study 2 (adults) we computed the area under the empirical discounting curve ($AUC$) (Note, due to the low number of varying rewards [only four different SS rewards], computing the area under the points of indifference would decrease variability and in consequence information when applied to the data in study 1). In detail, the $AUC$ corresponds to the area under the connected data points that describe the decrease of the subjective value (y-axis) over time (x-axis). Each specific delay was expressed as a proportion of the maximum delay and plotted against the normalized subjective (discounted) value. We then computed the area of the resulting trapezoids using Eq 1.

$$\frac{x_2 - x_1}{\left(\frac{(y_1 + y_2)}{2}\right)} \tag{Eq 1}$$

Smaller $AUC$-values indicate more discounting (more impulsive choices) and higher $AUC$-values indicate less discounting.

**Computational modeling.**   Based on prior analysis and basic research in the field of temporal discounting we a-priori assumed a hyperbolic model [63, 64] to model the decrease in subjective value over time. Bayesian estimation methods have the advantage of estimating the entire posterior distribution of parameter values given the data. Furthermore, hierarchical Bayesian parameter estimation benefits from the fact that the entire data set is taken into account via the hierarchical structure of the model. Parameters from each participant thus mutually inform and constrain each other (partial pooling), such that meaningful estimates can be derived even with limited data per subject (for details on Bayesian group comparison see [65, 66]; or for an overview see [67]. Due to the different time-scales of both intertemporal choice tasks in adolescents and adults we decided to compare hyperbolic (Eq 2) and exponential discounting (Eq 3) models. Both models assume that the LL reward, delivered after a specific delay (D), is devaluated via a subject specific discount rate ($k$) that weights the influence of time on subjective value (SV). A lower $k$-parameter reflects a lower weight on delay (reduced discounting) whereas a higher $k$-parameter reflects steeper discounting. Both models differ in the way they model this weight. In hyperbolic discounting the near future is discounted more heavily than distant events. In exponential discounting the discount rate is constant.

$$SV = \frac{LL}{(1 + kD)} \tag{Eq 2}$$

$$SV = LL * \exp(-kD) \tag{Eq 3}$$

After devaluating the delayed option a sigmoid function (softmax choice rule; Eq 4) maps the comparison of both the devaluated LL and SS option to choice probability on a trial by trial basis. Here a free $\beta$ inverse temperature parameter scales the influence of value differences on

choice. A high $\beta$ value implies that participants decide purely on value differences whereas lower values indicates higher choice stochasticity. For limit of $\beta = 0$ choices are completely random.

$$P(LL) = \frac{\exp(\beta * SV(LL))}{\exp(\beta * SV(SS)) + \exp(\beta * SV(LL))} \quad (Eq\ 4)$$

Models were fit using a hierarchical Bayesian framework to estimate parameter distributions via Markov Chain Monte Carlo (MCMC) sampling with JAGS [68]. Single subject parameters were drawn from group-level normal distributions, with mean and variance hyper-parameters that were themselves estimated from the data. Model convergence was assessed via the $\hat{R}$ -statistic (Gelman-Rubinstein convergence diagnostic) where values < 1.01. (two chains) were considered acceptable. For information on prior specification see **S1 Table**.

**Analyses of group differences.** Group comparisons were conducted by examining the differences in posterior distributions per parameter of interest (discount-rate $k$ and softmax $\beta$). For group comparisons, we report Bayes factors (directional Bayes Factor (dBF)) for directional effects for the hyperparameter difference distributions of patients with TS and controls. BFs were estimated via kernel density estimation using R (4.03) via the RStudio (1.3.1) interface. These are computed as the ratio of the integral of the posterior difference distribution from 0 to $\infty$ versus the integral from 0 to $-\infty$. Using common criteria [69], we considered BFs between 1 and 3 as anecdotal evidence, BFs > 3 as moderate evidence, and BFs > 10 as strong evidence. BFs > 30 and > 100 were considered as very strong and extreme evidence, respectively, the inverse of these reflect evidence in favor of the opposite hypothesis.

## Results

### Study 1

**Demographic characteristics and clinical assessment.** Demographic and clinical characteristics between adolescents with TS and controls are shown in **Table 1**. For demographic,

Table 1. Demographic, clinical and neuropsychological characteristics of adolescents with TS and healthy controls.

| | Adolescents with TS ($n$ = 19) | | Controls ($n$ = 19) | | $T/U/X^2$ | $p$ |
|---|---|---|---|---|---|---|
| | Mean | SD | Mean | SD | | |
| Age (Years) [a] | 14.21 | 2.37 | 14.21 | 2.53 | 0.000 | 1.000 |
| Male/Female [c] | 13/6 | - | 78.9 | - | 0.543 | 0.467 |
| Right-handed [c] | 14/19 | - | 84.2 | - | 1.276 | 0.435 |
| Current medication | 2/19 | - | - | - | - | - |
| YGTSS impairment | 16.00 | 8.00 | - | - | - | - |
| YGTSS | 23.37 | 12.38 | - | - | - | - |
| PUTS | 19.53 | 5.61 | - | - | - | - |
| FBB-ADHD[b] | 0.38 | 0.26 | 0.82 | 0.48 | -3.226 | 0.093 |
| SBB-ADHD[a] | 0.39 | 0.22 | 0.68 | 0.39 | 88.0 | 0.497 |
| CY-BOCS [b] | 6.84 | 6.31 | 0.08 | 0.277 | 21.50 | <0.001 |

ADHD, attention deficit hyperactivity disorder; CY-BOCS, Children's Yale-Brown Obsessive-Compulsive Scale; (FBB)-ADHD/(SBB)-ADHD, Fremdbeurteilungsbogen/Selbstbeurteilungsbogen für Aufmerksamkeitsdefizit-/Hyperaktivitätsstörungen; PUTS, Premonitory Urge for Tics Scale; TS, Tourette syndrome; YGTSS, Yale Global Tic Severity Scale.

a. T-test was used because data was normally distributed.

b. Mann Whitney U test was used because data was not normally distributed.

c. $X^2$ square test.

**Table 2. Model comparison of two variants of intertemporal choice.**

|  | Adolescents | | | Adults | | |
| --- | --- | --- | --- | --- | --- | --- |
|  | Patients with TS | Controls | Full model | Patients with TS | Controls | Full model |
| Hyberbolic | 791.5 | 878.8 | 1668.83 | 2538.4* | 2156.4* | 4701.7* |
| Exponential | 686.5* | 806.2* | 1535.92* | 2634.8 | 2297.8 | 4926.9 |

clinical and neuropsychological characteristics of adolescents with TS and controls adjusted for multiple comparison see **S2 Table**.

**Model free analysis.** Controls chose the LL option in 48.3% of all cases whereas patients with TS chose that option around 10% more often in 58.4% of all cases (see **S2 Fig**). Before using a parametric-t-test, we applied an arcsin-transformation on all mean choice proportions per participant. The groups did not differ significantly in the frequency of LL choices ($t_{(35.83)}$ = 1.0646; $p$ = 0.29).

**Computational modeling.** Model comparison via DIC [70] revealed a better fit of the exponential model (see **Table 2**). This holds when applying a full model including all participants from both groups or when modeling both groups separately (see **Table 2**). We next examined overall group differences for the discount-rate $k$ (**Fig 1A**). Analyzing the posterior group difference distribution (**Fig 1B**) revealed that 93% of the posterior distribution of controls is below the distribution of patients with TS. We then computed a dBF(see methods section) which quantifies the relative evidence for increases vs. decreases in patients compared to controls. This yielded a dBF of 12.52, i.e. given the data and model, an increase in discounting on the group level in controls was 12.52 times more likely than a decrease. The corresponding analysis of choice stochasticity is provided in **S3 Fig**.

Model comparison was based on the Deviance Information Criterion (DIC) [66] where lower values indicate a better model fit. The adolescent data were better accounted by a model

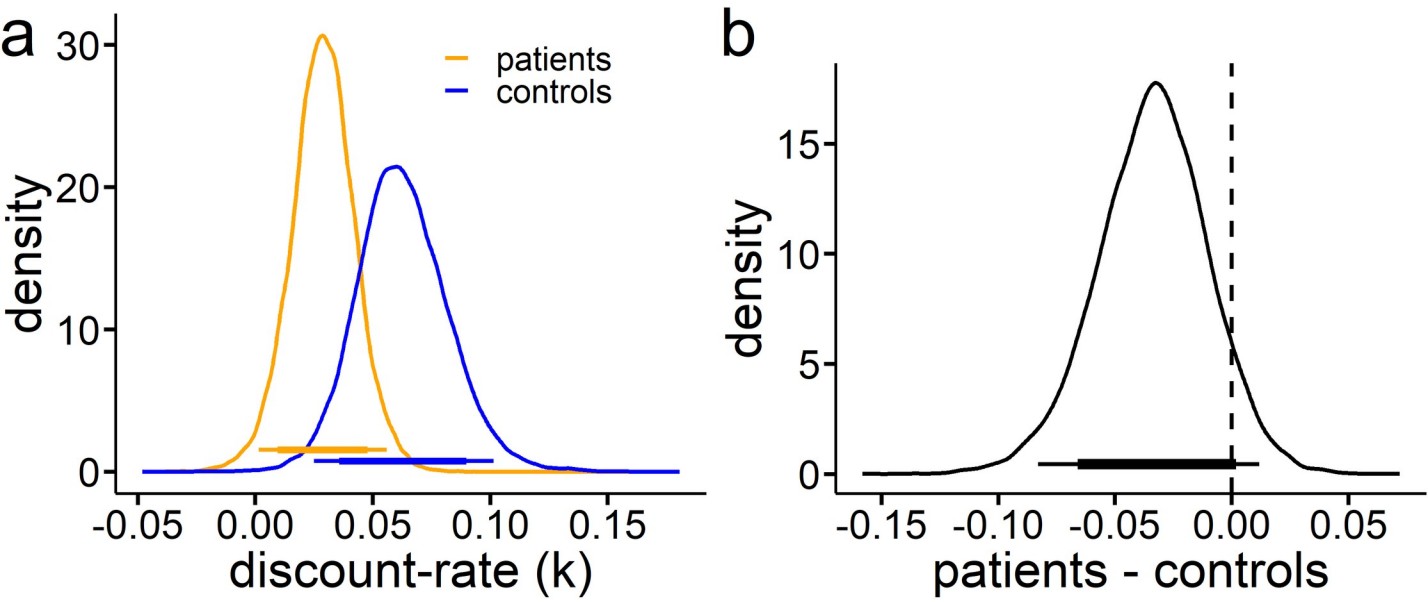

**Fig 1. a, Group level hyperparameter distributions of the discount-rate parameter k revealed that discounting was lower in adolescents with TS (orange) when compared to controls (blue). b, Difference distribution of controls—adolescence with TS.** Bayes factor for directional effects (dBF) indicated that a decrease in discount-rate ($k$) in patients was 12.52 times more likely than an increase. Thin and thick colored (a) and black (b) bars indicate the 95% and 85% highest density intervals respectively. TS, Tourette syndrome.

with an exponential discount function and the adult data were generally better accounted for by a temporal discounting model with hyperbolic discounting whereas.

## Study 2

**Demographic characteristics and clinical assessment.** Demographic and clinical characteristics of adult patients with TS and controls are shown in **Table 3**. Controls did not score in a clinically relevant range. Neither patients nor controls reported clinically relevant drug or alcohol abuse. We further conducted an analysis of correlations of discount-rate, age, compulsivity and symptom severity (**S3 Table**).

**Model free analysis.** Applying a parametric t-test on the integral of the area under the empirical discounting curve revealed no significant differences between patients with TS (mean[$AUC$] = 0.459) and controls (mean[$AUC$] = 0.511) ($t_{(46.1)}$ = -0.8791; $df$ = 46.1; $p$ = 0.38), see **S4 Fig**.

**Computational modeling.** Comparing hyperbolic and exponential discount functions based on the DIC [70] revealed a better fit of the hyperbolic model. This holds when applying a full model including all participants from both groups or when modeling both groups separately (see **Table 2**). In line with our model-agnostic approach, we did not find evidence for group differences when analyzing the posterior difference distribution of the discount-rate (k). Results are plotted in **Fig 2**. There was no evidence for consistent group differences (dBF = 0.38). The analysis was repeated excluding six patients with TS, that were taking antidopaminergic drugs at the time of the study. The exclusion of these patients only had a marginal effect and the result pattern did not change. For analysis of choice stochasticity see **S5 Fig**.

**Table 3. Demographic, clinical and neuropsychological characteristics of patients with TS and healthy controls.**

| | Adult patients with TS ($n$ = 25) | | Controls ($n$ = 25) | | $T/U/X^2$ | $p$ |
|---|---|---|---|---|---|---|
| | Mean | SD | Mean | SD | | |
| Age (Years)[a] | 29.88 | 9.03 | 29.40 | 9.28 | 0.185 | 0.854 |
| Male/Female [c] | 19/6 | - | 68.00 | - | 0.397 | 0.529 |
| Right-handed | 22/25 | - | 88.00 | - | 0.000 | 1.000 |
| Current medication | 9/25 | - | - | - | - | - |
| Years of education [b] | 11.68 | 1.25 | 11.90 | 1.22 | 250.00 | 0.197 |
| Tourette Onset | 8.76 | 5.13 | - | - | - | - |
| YGTSS motor | 15.84 | 5.72 | - | - | - | - |
| YGTSS verbal | 12.32 | 6.36 | - | - | - | - |
| YGTSS impairment | 26.80 | 11.08 | - | - | - | - |
| YGTSS | 54.96 | 20.78 | - | - | - | - |
| PUTS | 30.02 | 4.22 | - | - | - | - |
| BDI [b] | 11.68 | 9.34 | 5.28 | 5.19 | 165.50 | 0.004 |
| WURS-k [a] | 26.12 | 11.60 | 16.04 | 9.55 | 3.36 | 0.002 |
| OCI-R [b] | 20.30 | 12.06 | 10.92 | 7.58 | 149.50 | 0.002 |
| LPS-3 [b] | 55.80 | 8.25 | 58.60 | 8.48 | 249.50 | 0.213 |

BDI, Becks depression inventory; LPS-3, Leistungsprüfsystem; OCI-R, Obsessive-Compulsive Inventory-Revised; PUTS, premonitory urge tic for scale; TS, Tourette syndrome; WURS-k, Wender-Utah-Rating-Scale; YGTSS, Yale Global Tic Severity Scale.

a. T-test was used because data was normally distributed.

b. Mann Whitney U test was used because data was not normally distributed.

c. $X^2$ square test.

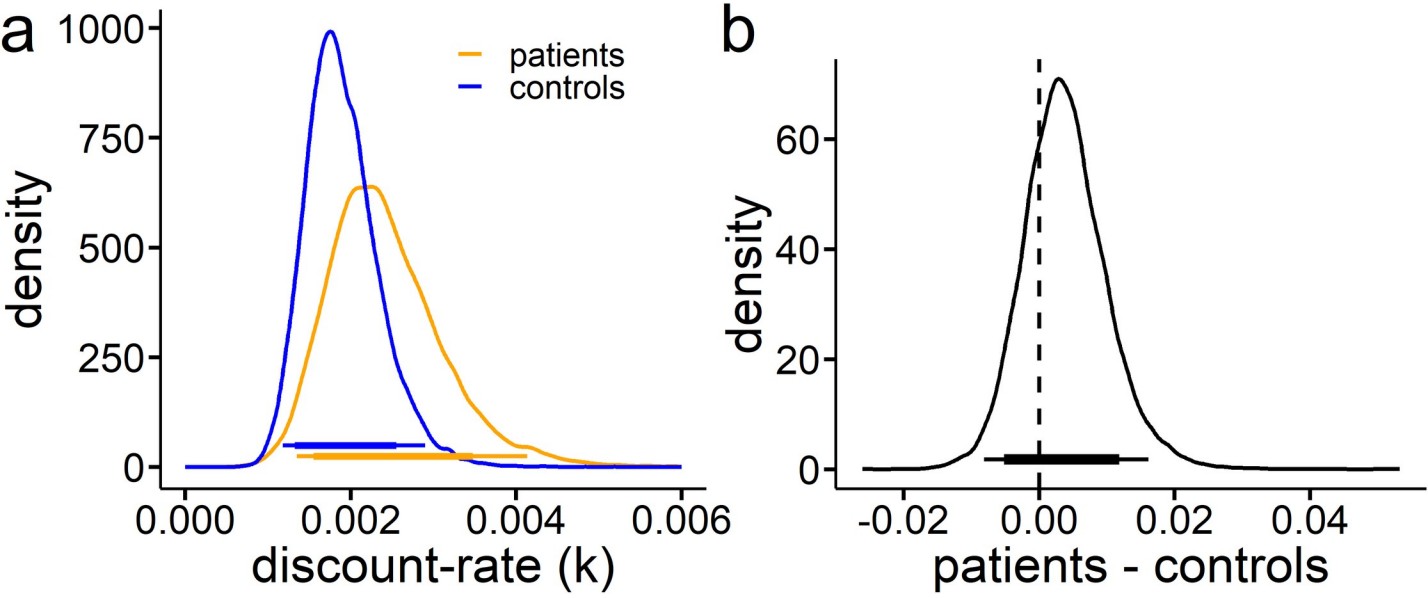

**Fig 2. a, Group level hyperparameter distributions of discount-rate *k* for patients with TS (orange) and controls (blue); b Difference distribution of patients with TS—controls.** The black bars indicate the 95% and 85% highest density interval respectively. Bayes factors for directional effects (dBF) of 0.36 patients > controls indicate no substantial difference between patients and controls. Thin and thick colored (a) and black (b) bars indicate the 95% and 85% highest density intervals respectively. TS, Tourette syndrome.

## Conclusions

The present study assessed temporal discounting in adolescent and adult patients with TS and matched healthy controls. Our data suggest reduced discounting (experiential task) in adolescent TS patients where in decrease in discounting was 12.52 times more likely than an increase when contrasted to controls. We did not find any difference in intertemporal choice in adults (hypothetical intertemporal choice task). TS is a complex neuropsychiatric disorder associated with developmental disturbances in dopaminergic transmission which possibly result in failure to control motor output [1, 2, 14, 15, 71]. These dopaminergic anomalies may either cause, enable or enhance tics via inadequate gating of information through the striatum [7]. Some studies point towards reductions in temporal discounting due to pharmacological elevation of DA levels, whereas others point to an increase [18]. Generally, the human literature on dopaminergic contributions to impulsivity is characterized by substantial heterogeneity [72]. A further complicating factor is that dopaminergic effects might be non-linear [73], as summarized in the inverted U-model of DA functioning [74]. However, acute dopaminergic modulation by pharmacological agents and long-term abnormal dopaminergic states such as in TS may effect behavior differently. In line with this distinction, our results suggest that the putative chronic hyperdopaminergic state of TS does not give rise to substantial changes in temporal discounting in adults.

However, we did find evidence for decrease in temporal discounting in adolescents with TS when compared to healthy controls, i.e. adolescents with TS preferred LL rewards.

Our analysis revealed that a decrease in temporal discounting in adolescents with TS was about 12 times more likely than an increase (dBF = 12.52). Adolescents typically show higher discount rates than adults [75]. This is thought to be attributable to functional and structural fronto-subcortical connectivity that undergoes maturation until early adulthood [15, 43, 44]. Adolescents with TS are constantly faced by tics and the need to control their motor output.

Even though these tics might emerge from complex neurophysiological interactions, i.e. hyper-active DA modulated striatal gating and reduced inhibition of GABAergic interneurons [76, 77], one could speculate that the ability to inhibit tics might foster the ability to inhibit other impulses, thereby strengthening cognitive control more generally [46]. However our results conflict with a recent study by Vicario and colleagues [27] who report increased discounting in adolescent patients with TS. We note that the task for the adolescent sample in our study differed distinctively, not only from the task of the adult sample but also from the Monetary Choice Questionnaire used by Vicario and colleagues [27]. Importantly, our task included a payout dependent on actual choice behavior (experiential task). Differences in the reported findings on impulsive choice of Vicario and colleagues and our findings might therefore be reflected in differences in task demands. In theory three complementary systems are thought to orchestrate intertemporal decisions: the valuation network, regions associated with cognitive control [40, 41], and a network associated with future prospection [29]. We therefore further propose that the weights between brain circuits involved in intertemporal choices might differ. That is the networks involved might depend on the temporal horizon of the task i.e. the need for future prospection might be less pronounced in the experiential task. However future studies are needed to clarify these issues.

The question then arises why such an effect would not likewise translate into greater self-control during temporal discounting in the adult TS patients. One possibility is that such a "training" account merely affects the developmental trajectory of self-control, such that adolescents with TS reach adult levels of self-control earlier than their healthy peers. Testing such a model would require longitudinal studies.

Additional clinical differences between adolescent and adult TS patients further complicate the interpretation of the differential effects in the two age groups. Adolescents and adults with TS exhibit different tic-phenomenology, for instance adolescents exhibit higher variability and/or fluctuations in tics. In consequence adolescents who successfully control their tics have a greater likelihood of eventual remission, likely due to better executive control capabilities [78]. In contrast, patients who still exhibit TS in adulthood exhibit attenuated inhibitory control [14]. In both samples, the discount rate ($k$) was not significantly correlated (corrected for multiple comparisons) with ADHD, OCD comorbid symptomatology or the YGTSS (see **S1** and **S2** **Tables**). Interestingly, the data in adolescent patients with TS was best fit by an exponential function, while the data in adult patients with TS was best accounted for by a model with hyperbolic discount function, which is in line with most of the literature on intertemporal choice [63]. First, though speculative the function of temporal discounting, processed in CSTC-loops, might generally be sensitive to the time scale (seconds/minute in adolescents vs. days to weeks in adults) of the task (see discussion of task differences above). Second, there might be technical reasons for this finding so the differences in the relative model fit between tasks could be due to differences in the option sets.

The present study has several limitations. First, adolescents and adults performed different temporal discounting tasks with different reward magnitudes (0–4 cents vs. 20–80 €) on a different timescale. Reward magnitudes in the range of cents vs. tens of Euros may entail different valuation and/or control processes [79, 80]. This precludes direct comparisons in $k$ between age groups. Importantly, both tasks experiential and hypothetical differ in what is known regarding their internal and external validity. While the hypothetical intertemporal choice task was proposed to constitute a transdiagnostic trait [81] less is known about the experiential task. Nonetheless, we note that the experiential task in study 1 is comparable with tasks like those used in the Marshmallow experiments by Mischel and Ebbesen [82] or other experiential adaptions [83, 84]. These experiential tasks have also shown some predictive value [85, 86] and successful treatment interventions in populations that are known for steep discounting [87].

Some studies do report correlation of experiential and hypothetical tasks (e.g. [88]). However, these findings are not always present [89, 90] and therefore represent a limitation of the current study.

Second, we draw theoretical conclusions from reward impulsivity to motor inhibition in patients with TS, even though motor inhibition was nor directly tested in the present studies. Further studies should further examine the developmental trajectories of both functions. Third, although only two adolescents with TS took medication, about a quarter of the adult patients ($n$ = 6) were on antidopaminergic medication. An integrative review showed that most TS medication (i.e. $D_2$ antagonists) reduce phasic DA, tonic DA or both [71] such that processing in fronto-striatal circuits was likely affected by the medication. However, a control analysis, excluding participants on antidopaminergic medication yielded the same pattern of results. Fourth, the samples may not be representative of the true TS population. Generalizability is limited due to the respective age ranges, the exclusion of patients with severe comorbidities and the fact that all patients were seeking treatment in a specialized outpatient clinic. Fifth, another limitation is the relatively small sample size of both studies. This is especially relevant for the interpretation of study 2, were no significant between-group differences were observed. Importantly, the lack of difference should be interpreted carefully with further studies needed to verify this finding.

The present study assessed temporal discounting in adolescent and adult patients with TS and matched healthy controls. Our data suggest reduced discounting (via an experiential task) in adolescent TS patients. We speculate that this might be due to improved inhibitory functions that affect choice impulsivity and/or the developmental trajectory of executive control functions. Interestingly, adult patients with TS exhibited levels of discounting similar to controls. This might be due to higher disease severity in adult patients with TS (e.g., patients who acquired successful tic inhibition during adolescence might have gone into remission). Future studies would benefit from adopting a consistent longitudinal approach to further elucidate the developmental trajectory of neural correlates i.e. dopaminergic states and intertemporal preferences and further from directly examining effects of dopaminergic medication on these processes in TS.

## Supporting information

**S1 Fig. Example for two trials in the temporal discounting task adapted for children and adolescents.** The blue circle depicts the LL reward (in cents) that participants will receive if they wait. How long they have to wait is indicated by blue lines, i.e. one blue line = 10s wait, six blue lines = 60s wait. The red circle indicates how much the participant will receive if they move on to the next trial immediately (0–4 cents). Participants received feedback about the amount earned after every trial (piggy bank). The green bar below the two circles indicates how many trials the participant has already finished. LL, larger but later.
(TIF)

**S2 Fig. Percentage of larger, but later choices in adolescents with TS and controls.** LL, larger but later.
(TIF)

**S3 Fig. A: Softmax $\beta$ in adolescent patients with TS vs. controls.** Group level hyperparameter distributions of the inverse temperature parameter softmax $\beta$ revealed no group differences between patients (orange) and controls (blue). **B: Difference distribution of controls—patients with TS.** Thin and thick colored (a) and black (b) bars indicate the 95% and 85%. TS, Tourette syndrome.
(TIF)

**S4 Fig. Subject specific comparison of the integral under the empirical area under the curve in adults with TS and controls.** TS, Tourette syndrome.
(TIF)

**S5 Fig. A: Softmax β in adult patients with TS vs. controls.** Group level hyperparameter distributions of the inverse temperature parameter softmax β revealed no group differences in the mean of the posterior or a shift in either direction between patients (orange) and controls (blue). However, variance was increased in controls indicating higher interindividual differences in decision noise. **B: Difference distribution of controls—patients with TS.** Thin and thick colored (a) and black (b) bars indicate the 95% and 85% highest density intervals respectively. TS, Tourette syndrome.
(TIF)

**S1 Table. Prior specifications for group and subject level parameters.** Discount-rate (k) parameters were estimated in logarithmic space due to parameter stability. Softmax β values were estimated in standard-normal space for the same reason.
(DOCX)

**S2 Table. Correlation analysis of model parameters and subscale of the SBB-Questionnaire adjusted for multiple comparison.** We report our exploratory analysis on discount-rate and questionnaire data. Scores are spearman correlation coefficients (p-value) not corrected for multiple comparisons. TS, Tourette syndrome.
(DOCX)

**S3 Table. Correlation analysis of model parameters and questionnaire data in adult patients with TS and controls adjusted for multiple comparison.** We report our exploratory analysis on discount-rate and questionnaire data. Scores are spearman correlation coefficients (p-value) not corrected for multiple comparisons. TS, Tourette syndrome; BDI, Becks depression inventory; OCI-R, Obsessive-Compulsive Inventory-Revised; TS, Tourette syndrome; WURS-k, Wender-Utah-Rating-Scale.
(DOCX)

## Author Contributions

**Conceptualization:** Canan Beate Schüller, Thomas Schüller, Julia Kerner auch Koerner, Alexander Münchau, Valerie Brandt, Jan Peters, Jens Kuhn.

**Data curation:** Canan Beate Schüller, Ben Jonathan Wagner, Julia Kerner auch Koerner, Eva Niessen.

**Formal analysis:** Canan Beate Schüller, Ben Jonathan Wagner, Jan Peters.

**Funding acquisition:** Canan Beate Schüller, Alexander Münchau, Valerie Brandt, Jens Kuhn.

**Investigation:** Canan Beate Schüller, Ben Jonathan Wagner, Valerie Brandt.

**Methodology:** Canan Beate Schüller, Ben Jonathan Wagner, Thomas Schüller, Valerie Brandt, Jan Peters, Jens Kuhn.

**Project administration:** Canan Beate Schüller, Eva Niessen, Valerie Brandt, Jens Kuhn.

**Software:** Ben Jonathan Wagner, Jan Peters.

**Supervision:** Thomas Schüller, Juan Carlos Baldermann, Daniel Huys, Alexander Münchau, Valerie Brandt, Jan Peters, Jens Kuhn.

**Writing – original draft:** Canan Beate Schüller, Ben Jonathan Wagner.

**Writing – review & editing:** Canan Beate Schüller, Ben Jonathan Wagner, Thomas Schüller, Juan Carlos Baldermann, Daniel Huys, Julia Kerner auch Koerner, Eva Niessen, Alexander Münchau, Valerie Brandt, Jan Peters, Jens Kuhn.

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
