## [Decision Letter · Decision Letter 0]

8 Apr 2021

PONE-D-21-03304

Temporal discounting in adolescents and adults with Tourette syndrome

PLOS ONE

Dear Dr. Schüller,

Thank you for submitting your manuscript to PLOS ONE. After careful consideration, we feel that it has merit but does not fully meet PLOS ONE’s publication criteria as it currently stands. Therefore, we invite you to submit a revised version of the manuscript that addresses the points raised during the review process.

I was able to obtain one review from an expert in the field.  The Reviewer noted several points of concern that should be addressed, and I agree with the points raised, based on my own reading of your manuscript.  Therefore, I invite you to submit a revision if you feel you can adequately address the concerns raised by the reviewer.  

We look forward to receiving your revised manuscript.

Kind regards,

Darrell A. Worthy, Ph.D

Academic Editor

PLOS ONE

Journal Requirements:

Reviewers' comments:

Reviewer's Responses to Questions

**Comments to the Author**

1. Is the manuscript technically sound, and do the data support the conclusions?

Reviewer #1: Yes

2. Has the statistical analysis been performed appropriately and rigorously? 

Reviewer #1: Yes

3. Have the authors made all data underlying the findings in their manuscript fully available?

Reviewer #1: No

4. Is the manuscript presented in an intelligible fashion and written in standard English?

Reviewer #1: Yes

5. Review Comments to the Author

Reviewer #1: This is a manuscript summarizing two studies investigating temporal discounting in people with Tourette's Syndrome (TS). The first study compared adolescent individuals with TS compared to age, education, and gender-matched controls and the second study compared adult individuals with TS compared to age, education, and gender-matched controls. These studies bring valuable information because there are few investigations of choice impulsivity in this clinical population and because current theories on the pathophysiological role of Dopamine alterations in TS suggest choice impulsivity might be affected.

Interestingly, the authors find that adults with TS are no different than controls in a standard incentive-compatible delay discounting task, and even more perplexingly, adolescents with TS are less impatient than controls. I absolutely think these results should be published but I have a few concerns that I would like the authors to address:

1. The manuscript does not cite a recent paper on delay discounting in TS (see Vicario et al. Mov Disord. 2020 Jul;35(7):1279-1280). This is a relevant study to include in the references and should definitely be contrasted with the results in study 1 because in that paper the finding goes in the opposite direction, where adolescents with TS were found to be steeper discounters than age-matched controls.

2. One issue is that for both study 1 and 2 sample size is small. Could the authors address whether the sample size was calculated and determined ex ante?

3. Could they discuss how representative is their sample of the real TS population?

4. How were the participants recruited? That is, were they enrolled from a treatment facility or from the community?

5. Could the authors discuss how the bayesian estimation method they applied impacts their ability to detect differences in the parameter distributions while having few observations?

6. What was the age range for the adolescent sample? Was there an effect of age on discounting in the patient and or control groups?

7. As the authors acknowledge, the experiential task used in study 1 is quite different from the more 'standard' delay discounting task used in study 2, which makes drawing conclusions about differences between adults with TS and adolescents with TS quite difficult. In fact, it has been shown that even subtle differences in the way that the SAME delay discounting task is presented (e.g. which of the options varies and which is fixed) can influence individual's behavior, so having participants perform DIFFERENT tasks is really almost like comparing apples to oranges. While the delay discounting task used in study 2 has had ample internal and external validation for its psychometric properties, less is known about the task employed in study 2 and the extent to which behavior on that task is predictive of any real-world behaviors. I think the manuscript should make this caveat more explicit in their discussion, specifically in the first and second paragraph of page 14 where they speculate about what could explain the different results found in both populations.

8. Another factor that may be hugely determinant to the parameters extracted from temporal discounting tasks is the choice of trials, that is, the set of options presented at each trial. How was the choice set determined for the experiential task? How precise can the estimated parameters be with your set of 50 trials? Were there enough trials on the impatient side of the choice set (it seems like the most 'impatient' trial was one where the choice was between no money now and 5 cents in 10 seconds) that would allow you to fit a hyperbolic model with precision?

9. Exponential models of discounting seem to fit better when the discount rates are not very high. Could the authors discuss whether the better fit could be a result of the choice set?

10. Establishing group differences (aka a diagnostic effect) is important and clearly the aim of your studies but did you explore the relationship between discounting and severity in your two studies?

11. As the authors point out choice impulsivity is distinct from motor impulsivity. Another, important distinction is the difference between impulsivity and compulsivity. Was there a relationship between discounting and compulsive symptomatology as assessed via CY-BOCS or the OCI-R?

12. The data from these studies will be useful for future studies looking at this clinical population. For this reason, one comment that is in keeping with the open science philosophy of this journal is that the data should be in a repository as opposed to available upon "reasonable" request.

13. Finally, a minor issue is that in the paragraph entitled "Analyses of group differences" in the Materials and Methods section, there seem to be some greater than symbols missing in the description of the interpretation of evidence provided by different levels of Bayes' factor.

6. PLOS authors have the option to publish the peer review history of their article (what does this mean?). If published, this will include your full peer review and any attached files.

Reviewer #1: No

---

## [Author Response · Author response to Decision Letter 0]

29 Apr 2021

Point-to-point responses:

Reviewer: 1

We thank the reviewer for the favorable, yet thorough evaluation of our manuscript and the helpful statements. 

Comment 1:

The manuscript does not cite a recent paper on delay discounting in TS (see Vicario et al. Mov Disord. 2020 Jul;35(7):1279-1280). This is a relevant study to include in the references and should definitely be contrasted with the results in study 1 because in that paper the finding goes in the opposite direction, where adolescents with TS were found to be steeper discounters than age-matched controls.

Response 1: 

We thank the reviewer for pointing our attention to this recent publication. As suggested, we now refer to the study by Vicario et al. (2020) in our introduction and discussion.

In the introduction, we now write (p. 4, line 87-90):

“However, the degree to which choice impulsivity is impaired in TS remains unclear. To date, only one study examined temporal discounting in patients with TS via the Kirby Monetary Choice Questionnaire [1] and observed slightly increased discounting in when compared to healthy controls.”

In the discussion, we now write (p.15, line 366-378): 

“However our results conflict with a recent study by Vicario and colleagues [1] who report increased discounting in adolescent patients with TS. We note that the task for the adolescent sample in our study differed distinctively, not only from the task of the adult sample but also from the Monetary Choice Questionnaire used by Vicario and colleagues [1]. Importantly, our task included a payout dependent on actual choice behavior (experiential task). Differences in the reported findings on impulsive choice of Vicario and colleagues [1] and our findings might therefore be reflected in differences in task demands. In theory three complementary systems are thought to orchestrate intertemporal decisions: the valuation network, regions associated with cognitive control [2,3], and a network associated with future prospection [29]. We therefore further propose that the weights between brain circuits involved in intertemporal choices might differ. That is the networks involved might depend on the temporal horizon of the task i.e. the need for future prospection might be less pronounced in the experiential task. However future studies are needed to clarify these issues.”

Comment 2: 

One issue is that for both study 1 and 2 sample size is small. Could the authors address whether the sample size was calculated and determined ex ante?

Response 2: 

Regarding the first study, the working group at the University of Cologne used an effect size of d = .7 and statistical power of .8 for between-group comparisons of 25 participants in each group. Due to the lack of studies on temporal discounting and Tourette´s Syndrome this effect size was based on studies investigating ADHD and temporal discounting [4–6].

For the adolescents, a power analysis was conducted for the ethics application. Because there were no previous studies on delay of gratification in adolescents with TS, we assumed a medium effect size, alpha = .05, and test power = 80, which results in a recruitment target of n = 20 patients with TS and n = 20 healthy controls. We managed to recruit n = 19 patients in the time-window of the study. 

We further added this point to the limitations section on p. 17, line 422-425

“Fifth, another limitation is the relatively small sample size of the studies. This is especially relevant for the interpretation of study 2, where no significant between-group differences were observed. Importantly, the lack of difference should be interpreted carefully with further studies needed to verify this finding.”

Comment 3:

Could they discuss how representative is their sample of the real TS population?

Response 3:

This is an interesting point. First, our samples do not represent the full age range observed in TS, with the sample of study 1 including only adolescent TS patients and the sample of study 2 including a wider age range from 19 till 53. Second, in adults TS is associated with comorbidities (most commonly ADHD, OCD and MDD) that often can be the primary reason for impaired quality of life. We aimed to include patients that are primarily affected by TS, albeit comorbid symptomatology was still present in both samples. Third, patients had to be affected in a way that made them seek treatment in a specialized outpatient clinic. Therefore, we do not claim that our sample is necessarily representative for the whole TS population.

We added the following point to our discussion on p. 17, line 419,422

“Fourth, the samples may not be representative of the true TS population. Generalizability is limited due to the respective age ranges, the exclusion of patients with severe comorbidities and the fact that all patients were seeking treatment in a specialized outpatient clinic.”

Comment 4: 

How were the participants recruited? That is, were they enrolled from a treatment facility or from the community?

Response 4: 

We thank Reviewer 1 for this valuable comment. We have added this information for adolescents and adults (p. 5, line 142-145 and p. 7, line 174-178), where we now write:

“Adolescents with TS were recruited from a specialist Tourette Syndrome clinic in Hamburg. All patients had been diagnosed with Tourette syndrome, some had been treated for their tics. Healthy controls were partly recruited from a pool of healthy participants who had participated in a previous study, partly via public advertisement.”

“We recruited twenty-five patients with diagnosed TS according to DSM-5 criteria [55]from the psychiatric outpatient clinic of the University Hospital Cologne (age: mean[sd] = 29.88 [±9.03]; 19 male subjects) and 25 age, education and gender-matched controls (age: mean[sd] = 29.40 [±9.28]; 17 male subjects) through public advertisement. 

Comment 5: 

Could the authors discuss how the bayesian estimation method they applied impacts their ability to detect differences in the parameter distributions while having few observations?

Response 5: 

We thank the reviewer for this important comment. We now further address the Bayesian estimation scheme in the methods section on p. 9 (line 224-230) and provide literature addressing the advantages of Bayesian estimation of group differences in detail.

“Bayesian estimation methods have the advantage of estimating the entire posterior distribution of parameter values given the data. Furthermore, hierarchical Bayesian parameter estimation benefits from the fact that the entire data set is taken into account via the hierarchical structure of the model. Parameters from each participant thus mutually inform and constrain each other (partial pooling), such that meaningful estimates can be derived even with limited data per subject (for details on Bayesian group comparison see [7,8];or for an overview see [9]).”

Comment 6:

What was the age range for the adolescent sample? Was there an effect of age on discounting in the patient and or control groups?

Response 6

We thank the reviewer for that point. We have now included the age range of both adults and adolescents (10-18 years for adolescents, and 19 to 53 for the adults). In adolescents (age range = 10-18), there is no significant association of age and discount-rate when pooling over all subjects (rho = -0.15, S = 10577, df = 36, p = 0.35). The correlation was numerically somewhat more pronounced in patients with TS (rho = -0.16, S = 1328.6, p = 0.5) compared to controls (rho = -0.09, S= 1244.6, p = 0.71) but non-significant. In adults (age range = 19-53), we found a significant negative correlation of age and discount-rate log(k) when pooling over all subjects (rho = –0.39, S = 28930, p = 0.005) indicating lower discounting with increasing age. The correlation was numerically similar in both groups (patients: rho = -0.39, S = 3625.8, p = 0.05; controls: rho = -0.40, S = 3691.6, p = 0.04).

We included these findings in the supporting information (S1 Table and S2 Table). 

Comment 7: As the authors acknowledge, the experiential task used in study 1 is quite different from the more 'standard' delay discounting task used in study 2, which makes drawing conclusions about differences between adults with TS and adolescents with TS quite difficult. In fact, it has been shown that even subtle differences in the way that the SAME delay discounting task is presented (e.g. which of the options varies and which is fixed) can influence individual's behavior, so having participants perform DIFFERENT tasks is really almost like comparing apples to oranges. While the delay discounting task used in study 2 has had ample internal and external validation for its psychometric properties, less is known about the task employed in study 2 and the extent to which behavior on that task is predictive of any real-world behaviors. I think the manuscript should make this caveat more explicit in their discussion, specifically in the first and second paragraph of page 14 where they speculate about what could explain the different results found in both populations.

Response 7: We followed the suggestion of the reviewer and have now included further discussion regarding the differences of both tasks and further emphasized the caveats between comparing those two tasks in the discussion section (p. 16, line 366-378 and p.17, line 402-410). 

“We note that the task for the adolescent sample in our study differed distinctively, not only from the task of the adult sample but also from the Monetary Choice Questionnaire used by Vicario and colleagues [1]. Importantly, our task included a payout dependent on actual choice behavior (experiential task). Differences in the reported findings on impulsive choice of Vicario and colleagues [1] and our findings might therefore be reflected in differences in task demands.”

“Importantly, both tasks experiential and hypothetical differ in what is known regarding their internal and external validity. While the hypothetical intertemporal choice task was proposed to constitute a transdiagnostic trait [10] less is known about the experiential task. Nonetheless, we note that the experiential task in study 1 is comparable with tasks like those used in the Marshmallow experiments by Mischel et al. [11] or other experiential adaptions [12,13]. These experiential tasks have also shown some predictive value [14,15] and successful treatment interventions in populations that are known for steep discounting [16]. Some studies do report correlation of experiential and hypothetical tasks (e.g. [17]). Yet these findings are not always present [18,19] and therefore represent a limitation of the current study.”

Comment 8: Another factor that may be hugely determinant to the parameters extracted from temporal discounting tasks is the choice of trials, that is, the set of options presented at each trial. How was the choice set determined for the experiential task? How precise can the estimated parameters be with your set of 50 trials? Were there enough trials on the impatient side of the choice set (it seems like the most 'impatient' trial was one where the choice was between no money now and 5 cents in 10 seconds) that would allow you to fit a hyperbolic model with precision?

Response 8: We thank the reviewer for this comment. Trials in this task were based on a diploma thesis. In this thesis this choice set showed high split-half reliability > 0.90 [20].

To further answer this question we extracted the estimated discount-rate parameters from both models (hyperbolic and exponential) to predict participants choices. We then computed an accuracy score, that is a proportion of how many of the choices have been correctly predicted given a model (hyperbolic or exponential) per participant. Overall, both the hyperbolic and exponential model each predict ~79% of all choices correctly (t = -0.50961, df = 37, p-value = 0.6133). We further elaborate on that in our answer to Comment #9. We also added another paragraph to our discussion (p.17, line 396-398) highlighting a possible association of choice-set and model-fit. 

“Second, there might be technical reasons for this finding so the differences in the relative model fit between tasks could be due to differences in the option sets.”

Comment 9: Exponential models of discounting seem to fit better when the discount rates are not very high. Could the authors discuss whether the better fit could be a result of the choice set?

Response 9: We believe that this point complements point 8) raised by the reviewer. We here performed one further analyses. We fitted both hyperbolic and exponential models to individual subjects, that is without the hierarchical structure. We then used the individual difference in model fit, that is the difference in DIC scores of each subject. That difference score is informative in terms of the difference in fit between both the exponential and hyperbolic model in each participant. We then performed a simple correlation analysis to see whether the general proportion of larger but later choices correlates with differences in model-fit (r = 0.19, t = 1.1402, df = 36, p-value = 0.2617). Even though the relationship was positive in nature, indicating the direction the reviewer mentions, the correlation was not statistically significant. 

We do therefore address this possibility in our discussion (p. 17, line 394-398). 

„ First, though speculative the function of temporal discounting, processed in CSTC-loops, might generally be sensitive to the time scale (seconds/minute in adolescents vs. days to weeks in adults) of the task (see discussion of task differences above). Second, there might be technical reasons for this finding so the differences in the relative model fit between tasks could be due to differences in the option sets.”

Comment 10: Establishing group differences (aka a diagnostic effect) is important and clearly the aim of your studies but did you explore the relationship between discounting and severity in your two studies?

Response 10: We followed the reviewers suggestion and ran exploratory analyses of the association between symptom severity (YBGTSS scores) and discount-rate (k). There were no significant associations (adolescents: rho = -0.03 =, S = 578.15, p = 0.91, adults: rho = 0.16, S = 2179.8 , p = 0.44). Four YBGTSS values were missing in the adolescent group. We added this analyses to our corresponding tables (S2 Table and S3 Table).

Comment 11: As the authors point out choice impulsivity is distinct from motor impulsivity. Another, important distinction is the difference between impulsivity and compulsivity. Was there a relationship between discounting and compulsive symptomatology as assessed via CY-BOCS or the OCI-R?

Response 11: Thank you for that valid point. We now report correlations with discounting and compulsive symptomatology in our corresponding Tables (S2 Table and S3 Table).

In adolescents, CY-BOCS scores were negatively related to discount-rate k on a trend level (rho = -0.39, S = 1585.3 , p = 0.098) implying decreased discounting with increased CY-BOCS score. In adults, we found no significant relationship of OCI-R and our discount-rate log(k) model parameters (rho = 0.17, S = 2149.5, p = 0.41).

Comment 12: The data from these studies will be useful for future studies looking at this clinical population. For this reason, one comment that is in keeping with the open science philosophy of this journal is that the data should be in a repository as opposed to available upon "reasonable" request

Response 12: We followed the reviewers suggestion and have now made the data publicly available on the platform: center for open science (Schüller C. Temporal discounting in adolescents and adults with Tourette syndrome [Internet]. OSF; 10 Apr 2021. Available: osf.io/nw9pc).

Comment 13: Finally, a minor issue is that in the paragraph entitled "Analyses of group differences" in the Materials and Methods section, there seem to be some greater than symbols missing in the description of the interpretation of evidence provided by different levels of Bayes' factor.

Response 13: We have now corrected that and added the missing signs in the “Analyses of group differences” paragraph on p. 10 (not marked).

References for the Rebuttal Letter:

1. Vicario CM, Gulisano M, Maugeri N, Rizzo R. Delay Reward Discounting in Adolescents With Tourette’s Syndrome. Mov Disord. 2020;35: 1279–1280. doi:10.1002/mds.28096

2. Figner B, Knoch D, Johnson EJ, Krosch AR, Lisanby SH, Fehr E, et al. Lateral prefrontal cortex and self-control in intertemporal choice. Nat Publ Gr. 2010;13: 538–539. doi:10.1038/nn.2516

3. Hare TA, Hakimi S, Rangel A. Activity in dlPFC and its effective connectivity to vmPFC are associated with temporal discounting. Front Neurosci. 2014;8: 1–15. doi:10.3389/fnins.2014.00050

4. Demurie E, Roeyers H, Baeyens D, Sonuga-Barke E. Temporal discounting of monetary rewards in children and adolescents with ADHD and autism spectrum disorders. Dev Sci. 2012;15: 791–800. doi:10.1111/j.1467-7687.2012.01178.x

5. Chantiluke K, Christakou A, Murphy CM, Giampietro V, Daly EM, Ecker C, et al. Disorder-specific functional abnormalities during temporal discounting in youth with Attention Deficit Hyperactivity Disorder ( ADHD ), Autism and comorbid ADHD and Autism. Psychiatry Res Neuroimaging. 2014;223: 113–120. doi:10.1016/j.pscychresns.2014.04.006

6. Paloyelis Y, Asherson P, Mehta MA, Faraone S V., Kuntsi J. DAT1 and COMT effects on delay discounting and trait impulsivity in male adolescents with attention deficit/hyperactivity disorder and healthy controls. Neuropsychopharmacology. 2010;35: 2414–2426. doi:10.1038/npp.2010.124

7. Kruschke JK. Bayesian Assessment of Null Values Via Parameter Estimation and Model Comparison. 2011. doi:10.1177/1745691611406925

8. Baldwin SA, Fellingham GW. Bayesian methods for the analysis of small sample multilevel data with a complex variance structure. Psychol Methods. 2013;18: 151–164. doi:10.1037/a0030642

9. Wagenmakers EJ, Marsman M, Jamil T, Ly A, Verhagen J, Love J, et al. Bayesian inference for psychology. Part I: Theoretical advantages and practical ramifications. Psychon Bull Rev. 2018;25: 35–57. doi:10.3758/s13423-017-1343-3

10. Lempert KM, Steinglass JE, Pinto A, Kable JW, Simpson HB. Can delay discounting deliver on the promise of RDoC? Psychol Med. 2019;49: 190–199. doi:10.1017/S0033291718001770

11. Mischel W, Ebbesen EB. Attention in delay of gratification. J Pers Soc Psychol. 1970;16: 329–337. doi:10.1037/h0029815

12. Johnson MW. An efficient operant choice procedure for assessing delay discounting in humans: Initial validation in cocaine-dependent and control individuals. Exp Clin Psychopharmacol. 2012;20: 191–204. doi:10.1037/a0027088

13. Jimura K, Myerson J, Hilgard J, Braver TS, Green L. Are people really more patient than other animals? Evidence from human discounting of real liquid rewards. Psychon Bull Rev. 2009;16: 1071–1075. doi:10.3758/PBR.16.6.1071

14. Mischel W, Shoda Y, Peake PK. The Nature of Adolescent Competencies Predicted by Preschool Delay of Gratification. J Pers Soc Psychol. 1988;54: 687–696. doi:10.1037/0022-3514.54.4.687

15. Mischel W, Shoda Y, Rodriguez MI. Delay of gratification in children. Science (80- ). 1989/05/26. 1989;244: 933–938. Available: http://www.ncbi.nlm.nih.gov/pubmed/2658056

16. Gawrilow C, Gollwitzer PM, Oettingen G. If-then plans benefit delay of gratification performance in children with and without ADHD. Cognit Ther Res. 2011;35: 442–455. doi:10.1007/s10608-010-9309-z

17. Steele CC, Gwinner M, Smith T, Young ME, Kirkpatrick K. Experience Matters: The Effects of Hypothetical versus Experiential Delays and Magnitudes on Impulsive Choice in Delay Discounting Tasks. Brain Sci. 2019;9: 379. doi:10.3390/brainsci9120379

18. Smits RR, Stein JS, Johnson PS, Odum AL, Madden GJ. Test-retest reliability and construct validity of the experiential discounting task. Exp Clin Psychopharmacol. 2013;21: 155–163. doi:10.1037/a0031725

19. Patt V, Hunsberger R, Jones D, Keane M, Verfaellie M. Temporal discounting when outcomes are experienced in the moment: Validation of a novel paradigm and comparison with a classic hypothetical intertemporal choice task. 2020 [cited 29 Apr 2021]. doi:10.31234/osf.io/5ajuk

20. Janßen H. Vergleichende Untersuchung der Zukunftsorientierung des Wahlverhaltens von Kindern mit und ohne Aufmerksamkeitsdefizit-/Hyperaktivitätsstörung in einem Belohnugsaufschub-Paradigma mit realen Temporal Discounting Design. 2011. 

Journal Requirement:

Comment 1. Please ensure that your manuscript meets PLOS ONE's style requirements, including those for file naming.

Response 1: We have changed the manuscript to the PLOS ONE’s style requirements (not marked) and file naming accordingly. 

Comment 2. We note that you have indicated that data from this study are available upon request. PLOS only allows data to be available upon request if there are legal or ethical restrictions on sharing data publicly. For information on unacceptable data access restrictions, please see http://journals.plos.org/plosone/s/data-availability#loc-unacceptable-data-access-restrictions.

Response 2: We have now made the data publicly available on the platform: center for open science (Schüller C. Temporal discounting in adolescents and adults with Tourette syndrome [Internet]. OSF; 10 Apr 2021. Available: osf.io/nw9pc).

Comment 3. PLOS requires an ORCID iD for the corresponding author in Editorial Manager on papers submitted after December 6th, 2016. Please ensure that you have an ORCID iD and that it is validated in Editorial Manager. To do this, go to ‘Update my Information’ (in the upper left-hand corner of the main menu), and click on the Fetch/Validate link next to the ORCID field. This will take you to the ORCID site and allow you to create a new iD or authenticate a pre-existing iD in Editorial Manager. Please see the following video for instructions on linking an ORCID iD to your Editorial Manager account: https://www.youtube.com/watch?v=_xcclfuvtxQ

Response 3: I have created an ORCID iD account and linked it to my PlosOne account.

Comment 4. Please include captions for your Supporting Information files at the end of your manuscript, and update any in-text citations to match accordingly. Please see our Supporting Information guidelines for more information: http://journals.plos.org/plosone/s/supporting-information

Response 4: We corrected this mistake and included the captions for our Supporting information at the end of our manuscript.

Comment 5: While revising your submission, please upload your figure files to the Preflight Analysis and Conversion Engine (PACE) digital diagnostic tool, https://pacev2.apexcovantage.com/. 

Response 5: We used the tool “PACE” and our figures now meet the PlosOne requirements.

Comment 6: PLOS authors have the option to publish the peer review history of their article (what does this mean?). If published, this will include your full peer review and any attached files.

Response 5: We are agree to have the peer review history published.

---

## [Decision Letter · Decision Letter 1]

9 Jun 2021

Temporal discounting in adolescents and adults with Tourette syndrome

PONE-D-21-03304R1

Dear Dr. Schüller,

We’re pleased to inform you that your manuscript has been judged scientifically suitable for publication and will be formally accepted for publication once it meets all outstanding technical requirements.

I sent your revised manuscript back to the original reviewer, and the reviewer felt you had adequately addressed all of their comments.  I agree with the reviewer's assessment.

Kind regards,

Darrell A. Worthy, Ph.D

Academic Editor

PLOS ONE

Additional Editor Comments (optional):

Reviewers' comments:

Reviewer's Responses to Questions

**Comments to the Author**

1. If the authors have adequately addressed your comments raised in a previous round of review and you feel that this manuscript is now acceptable for publication, you may indicate that here to bypass the “Comments to the Author” section, enter your conflict of interest statement in the “Confidential to Editor” section, and submit your "Accept" recommendation.

Reviewer #1: All comments have been addressed

2. Is the manuscript technically sound, and do the data support the conclusions?

Reviewer #1: Yes

3. Has the statistical analysis been performed appropriately and rigorously? 

Reviewer #1: Yes

4. Have the authors made all data underlying the findings in their manuscript fully available?

Reviewer #1: Yes

5. Is the manuscript presented in an intelligible fashion and written in standard English?

Reviewer #1: Yes

6. Review Comments to the Author

Reviewer #1: (No Response)

7. PLOS authors have the option to publish the peer review history of their article (what does this mean?). If published, this will include your full peer review and any attached files.

Reviewer #1: **Yes: **Silvia Lopez-Guzman

---

## [Editor Report · Acceptance letter]

11 Jun 2021

PONE-D-21-03304R1 

Temporal discounting in adolescents and adults with Tourette syndrome 

Dear Dr. Schüller:

I'm pleased to inform you that your manuscript has been deemed suitable for publication in PLOS ONE. Congratulations! Your manuscript is now with our production department. 

Kind regards, 

on behalf of

Dr. Darrell A. Worthy 

Academic Editor

PLOS ONE